# Endoscopic Management of Adenomas in the Ileal Pouch and the Rectal Remnant after Surgical Treatment in Familial Adenomatous Polyposis

**DOI:** 10.3390/jcm11123562

**Published:** 2022-06-20

**Authors:** Masahiro Tajika, Tsutomu Tanaka, Sachiyo Oonishi, Keisaku Yamada, Tomoyasu Kamiya, Nobumasa Mizuno, Takamichi Kuwahara, Nozomi Okuno, Shin Haba, Yasuhiro Kuraishi, Akira Ouchi, Yusuke Sato, Takashi Kinoshita, Koji Komori, Kazuo Hara, Waki Hosoda, Yasumasa Niwa

**Affiliations:** 1Department of Endoscopy, Aichi Cancer Center Hospital, Nagoya 464-8681, Japan; tstanaka@aichi-cc.jp (T.T.); soonishi@aichi-cc.jp (S.O.); k.yamada@aichi-cc.jp (K.Y.); mr.children_365days@docomo.ne.jp (T.K.); yniwa@aichi-cc.jp (Y.N.); 2Department of Gastroenterology, Aichi Cancer Center Hospital, Nagoya 464-8681, Japan; nobumasa@aichi-cc.jp (N.M.); kuwa_tak@aichi-cc.jp (T.K.); nokuno@aichi-cc.jp (N.O.); s.haba@aichi-cc.jp (S.H.); y.kuraishi@aichi-cc.jp (Y.K.); khara@aichi-cc.jp (K.H.); 3Department of Gastroenterological Surgery, Aichi Cancer Center Hospital, Nagoya 464-8681, Japan; akira.ouc@aichi-cc.jp (A.O.); yu.sato@aichi-cc.jp (Y.S.); t-kinoshita@aichi-cc.jp (T.K.); kkomori@aichi-cc.jp (K.K.); 4Department of Pathology and Molecular Diagnostics, Aichi Cancer Center Hospital, Nagoya 464-8681, Japan; whosoda@aichi-cc.jp

**Keywords:** familial adenomatous polyposis, ieal pouch, endoscopic surveillance, endoscopic treatment, argon plasma coagulation

## Abstract

In patients with familial adenomatous polyposis (FAP), adenomas and even carcinomas may develop in the rectal remnant and the ileal pouch after surgical treatment. The aim of this study was to evaluate the outcome of endoscopic management in patients with FAP. The main outcome measurements were the appearance of secondary cancer, complications, and the need for additional surgery. Thirty-four FAP patients with Kock’s continent ileostomy (Kock) (*n* = 3), ileorectal anastomosis (IRA) (*n* = 12), and ileal pouch-anal anastomosis (IPAA) (*n* = 19) were identified. The median follow-up period of endoscopic surveillance was 11.5 years for pouch patients (Kock + IPAA) and 21.7 years for IRA. Metachronous adenomas appeared in 32 patients (94.1%). In pouch patients, a total of 120 treatments were given to 20 patients, and 12 sessions of delayed bleeding (10%) occurred, which was significantly higher compared to IRA patients, with 0 sessions (*p* < 0.001). In IRA patients, a total of 169 treatments were given to 11 patients, with one case of perforation. No adenocarcinoma has developed since the start of endoscopic surveillance. Regular endoscopic surveillance and treatment are feasible and safe. However, in pouch patients, one must be cautious about delayed bleeding in the treatment of adenomas.

## 1. Introduction

A restorative proctocolectomy with an ileal pouch-anal anastomosis (IPAA) is the standard surgical treatment for patients with familial adenomatous polyposis (FAP) to prevent the development of colorectal cancer. Alternatively, colectomy with an ileorectal anastomosis (IRA) is widely accepted, especially for patients with a low risk of rectal cancer. The major advantage of IRA is the preservation of rectal innervation, which leads to a better quality of life. However, continuing endoscopic surveillance for adenomas in the rectum is necessary because the cumulative risk of secondary proctectomy due to rectal cancer or uncontrollable rectal polyposis was 30% to 53% 25–30 years after colectomy [1,2,3]. On the other hand, it has been thought that IPAA theoretically eliminates the risk of colorectal cancer and adenomas and the need for further lower gastrointestinal surveillance. However, since Beart et al. [4] first reported the development of multiple adenomatous polyps in an ileal pouch in a FAP patient with a Kock pouch continent ileostomy in 1982, more than 30 reports related to ileal pouch adenomas have been reported [5,6,7,8], and the necessity for endoscopic surveillance of the ileal pouch in patients with IPAA has been well recognized.

We previously reported that the overall risk of adenoma development was significantly higher in IRA patients over a median follow-up period of 21.0 years, with incidence rates of 85% and 100% at 5 and 10 years of follow-up, respectively, compared to pouch patients (*p* < 0.001) [9]. However, there was also a high frequency of adenomas in the ileal pouch, with rates of 12%, 33%, and 68%, at 5, 10, and 20 years of follow-up, respectively. Furthermore, we reported six cases of advanced adenomas, including two cases of adenocarcinomas, that developed in an ileal pouch [9]. Regarding cancer development, several FAP patients whose cancers developed in an ileal pouch have been reported [5,6,7,10,11]. There is almost no doubt that adenomas or carcinomas frequently develop not only in the rectal remnant after IRA but also in the ileal pouch after IPAA. Therefore, endoscopic surveillance is recommended for FAP patients after surgery, but the outcomes of endoscopic surveillance and treatment have not been well reported.

The aim of this study was to evaluate the feasibility and safety of endoscopic treatment for the rectum and the ileal pouch in patients with FAP after surgery.

## 2. Materials and Methods

The endoscopic and medical records of all of the patients with FAP treated in Aichi Cancer Center Hospital (ACCH), Nagoya, Japan, between January 1965 and December 2019 were reviewed. In our hospital, endoscopic surveillance began in December 1981 for IRA patients and in April 2002 for pouch patients. FAP was defined by the presence of more than 100 colorectal adenomas and a family history of FAP. Of the patients who received surgical treatment such as IPAA, IRA, and Kock, patients who had a follow-up period of at least one year and at least one endoscopic examination during their follow-up were eligible for this study. The patients not participating in surveillance after surgery who visited our hospital for some complaint related to recurrence, such as melena, were excluded. In this study, the patients who had undergone pouch construction, either Kock or IPAA, were defined as pouch patients [12]. The main outcome measurements were the appearance of secondary cancer, the need for additional surgery, and complications.

In general, it was recommended that IRA patients have endoscopic examinations every 6 months and that pouch patients have annual examinations after surgery [13]. The interval between surgery and adenoma development was defined as the time from surgery to the first report showing histologically confirmed adenomas in the ileal mucosa. For each patient, the most advanced histological diagnosis was taken as valid.

The endoscopic surveillance protocol from 2001 was as follows. All of the patients were administered with 1 L of polyethylene glycol-electrolyte solution on the morning of the examination. Scopolamine butyl bromide (10 mg) or glucagon (0.5 mg) was administered intravenously in patients without contraindications to reduce bowel movements. A flexible endoscope (CF 200Z, CF Q240ZI, GIF H260, GIF H260Z, GIF H290Z; Olympus Optical Co. Ltd., Tokyo, Japan) was used for the examination. In addition to a thorough examination of the pouch in Kock or IPAA patients and rectum in IRA patients, the distal 25 to 30 cm of the afferent limb and the anal canal were also examined carefully. After a complete initial examination with white light imaging, chromoendoscopy using 0.2% indigo carmine solution was performed using a spray catheter (PW-5L-1; Olympus Optical Co. Ltd., Tokyo, Japan). If a polyp was detected, magnifying endoscopy with narrow-band imaging (NBI) or indigo carmine solution was performed to evaluate the polyp to determine if it was an adenoma using the JNET classification or pit pattern classification [14,15]. Advanced adenomas were defined as adenomas ≥10 mm in greatest diameter and/or with high-grade dysplasia. The sizes of polyps were estimated using the diameter of the endoscope, the width of biopsy forceps, or the spray catheter.

Endoscopic mucosal resection (EMR) or cold-snare polypectomy was performed for raised sessile lesions more than 6 mm in diameter (Figure 1). If the lesion was more than 2 cm, endoscopic piecemeal mucosal resection (EPMR) was performed (Figure 2). In cases of diffuse polyposis (more than 100 lesions) with large lesions that required EPMR, multistep treatment was performed at an interval of 3 months. Since 2004, Argon plasma coagulation (APC; the APC 300/Erbotom ICC200 or the APC2/VIO 300D, ERBE, Tübingen, Germany) has been performed for all confirmed small lesions less than 5 mm (Figure 3), and APC was also performed to treat residual lesions after EMR or EPMR. APC was performed using a coagulation mode at settings of 1.0 L per minute and 30 Watts.

The endoscopic surveillance protocol before 2001 was as follows. Bowel preparation was not strictly required; some cases underwent an enema prior to endoscopy, and the others were administered polyethylene glycol-electrolyte solution on the morning of the examination. There is no record of the endoscope, TCF-2L, CF 200Z, CF 240ZI, GIF XQ240, and GIF Q240; Olympus Optical Co. Ltd., Tokyo, Japan, might be used for the examination. In most cases, after an initial examination with white light imaging, chromoendoscopy using 0.2% indigo carmine solution was performed. In principle, the indication of EMR was an adenoma of more than 10 mm. All adenomas less than 10 mm in size, regardless of their number and shape, were coagulated. Before 2004, the Yd-YAG laser (Molectron Detector Inc. Portland, OR, USA) for rectal adenoma and the heater probe for ileal pouch adenoma were used.

Delayed bleeding was defined as melena which required additional treatment the day after the procedure.

This study was approved by the ethics committee of ACCH (ACC 2018-1-003), and all patients provided their informed consent for the collection and subsequent use of the data for research purposes. The study was carried out in accordance with the Helsinki Declaration.

### Statistical Analysis

Differences in proportions between patients were compared by Pearson’s chi-squared test. The Mann–Whitney U test was used to compare differences between medians. A *p*-value < 0.05 was considered significant. The data files were analyzed using JMP^®^ 12 (SAS Institute Inc., Cary, NC, USA).

## 3. Results

A total of 84 patients with FAP were operated on in our hospital between January 1965 and December 2019; of them, 22 pouch patients (11 males; median age at surgery: 30.6 years) and 12 IRA patients (4 males; median age at surgery: 36.7 years) were eligible for this study. Patient flow was shown in Figure 4. At our hospital, endoscopic surveillance began in December 1981 for IRA patients and in April 2002 for pouch patients.

Of the 84 patients, IPAA was performed in 28 patients, IRA was performed in 16 patients, and Kock was performed in eight patients. Of these patients, one died of pouch cancer, one died of gastric cancer, one died of acute leukemia, one died of pneumonia, and one died of food poisoning; they were all excluded from this study because they were not eligible. Among them, a patient with pouch cancer and a patient with acute leukemia had been detected in advanced pouch cancer, respectively. They were not participating in a surveillance program after surgery and visited our hospital with complaints such as anal bleeding, abdominal pain, or vomiting [10]. The former patient is the case that triggered us to start the surveillance of the IPAA patients in 2001. Table 1 shows the characteristics of the pouch patients and IRA patients. There were no significant differences in age at the time of surgery, sex, median polyp count at surgery, and co-existence of gastric polyps, papillary adenomas, or extra-papillary adenomas between pouch patients and IRA patients. There were significant differences in median duration to the first endoscopy after surgery and the first detection of adenoma in rectal mucosa and ileal-pouch mucosa after surgery between the two (both *p* < 0.001). There were no significant differences in adenoma development in the ileal pouch and rectal remnant.

The clinical outcomes of endoscopic surveillance and treatment are shown in Table 2. The median follow-up period from the start of endoscopic surveillance was 11.3 years in pouch patients and 21.7 years in rectal remnant patients. Although there were no significant differences in the number of endoscopic surveillance sessions, the number of treated adenomas per treatment, and the cumulative number of treated adenomas per patient, the number of endoscopic treatments in IRA patients showed a higher tendency compared to that in pouch patients (*p* = 0.074). Although APC was performed for most treatments, 96.7% of pouch patients and 95.6% of IRA patients, cold polypectomy was more common in pouch patients than in IRA patients. During the follow-up period, there was no development of adenocarcinomas and no need for additional surgical treatment in both groups. In complications, delayed bleeding was significantly more common in pouch patients than in IRA patients (*p* < 0.001). On the other hand, although there was no significant difference between the two groups, delayed perforation occurred in only one IRA patient. In this case, perforation occurred at the ileal mucosa connected to the rectum, not at the rectum; in the evening that the APC had been performed, the APC was performed for the adenomas appeared in the ileal mucosa.

## 4. Discussion

In FAP patients with total colectomy and IRA, there is almost no doubt that adenomas and adenocarcinomas develop in the rectal remnant. On the other hand, in patients with proctocolectomy and IPAA, the incidence of adenomas and adenocarcinoma development in the ileal pouch mucosa and the anal transitional zone in IPAA is lower than that in the rectum in IRA patients [9]. However, the incidence is not negligible. The European Society of Gastrointestinal Endoscopy (ESGE) recommends endoscopic removal of all polyps > 5 mm during surveillance of the rectal remnant or ileal pouch in patients with FAP in the ESGE guideline [16]. However, as mentioned in the guideline, the evidence on how to manage polyps in the rectal remnant or ileal pouch and the appropriate interval between endoscopies is scarce.

In the present retrospective study, the feasibility and safety of endoscopic treatment for the rectum and the ileal pouch were evaluated in patients with FAP after surgery. During median follow-up periods from the start of endoscopic surveillance of 11.3 years in pouch patients and 21.7 years in IRA patients, no adenocarcinoma developed in the rectal remnant or ileal pouch, and no additional surgical treatment was needed. In the present study, the small sample size was certainly a limitation, but this intensive endoscopic surveillance and treatment may reduce the development of adenocarcinomas and the need for additional surgical treatment. Ishikawa et al. reported that repeated endoscopies to remove numerous polyps in patients with FAP refusing colectomy were feasible and safe in the medium term [17]. In their study, the median numbers of endoscopic treatment sessions and polyps removed per patient were 8 and 475, respectively, for 90 patients with FAP during a median follow-up of 5.1 years. During this period, a total of more than 55,000 polyps were resected without adverse events such as perforation and bleeding, although five non-invasive carcinomas were detected within 10 months from the start of the follow-up period, which were treated endoscopically without recurrence. Only two patients (2.2%) underwent colectomy because the polyposis phenotype had changed to dense polyposis. In the present study, since the lengths of the rectal remnant and of the ileal pouch were short compared to the whole colon in the previous study, and the basic strategy for polyp management was different between the present study and the previous study, endoscopic treatment for the remnant intestine was strictly manageable and feasible.

Regarding the interval for lower endoscopy, Saurin et al. recommended an interval of 6 to 12 months in IRA patients and 1 to 2 years in pouch patients after surgery [11]. In the present study, the median interval of lower endoscopy was 6 months in IRA patients and 12 months in pouch patients, and the median number of treated adenomas at a time was 13.5 in IRA patients and 12.5 in pouch patients. Although the number of patients was very small, no secondary adenocarcinomas developed from the start of endoscopic surveillance. Pasquer et al. reported, in an analysis of 296 FAP patients (92 IPAA patients and 197 IRA patients), that aggressive and intense endoscopic treatment might decrease secondary cancer [18]. Their endoscopic follow-up regimen is recommended 6 months after surgery and then at least annually in cases of IRA and biennially in cases of IPAA. Endoscopic treatments such as APC, cold-snare biopsy, EMR, and submucosal dissection were performed for remaining or newly developed adenomas. During a median follow-up period of 17.1 years, secondary cancer occurred in 13 patients: 12 patients (6.1%) in the IRA group, 10 of whom were diagnosed at the first endoscopy in their hospital, and one patient (1.1%) in the IPAA group. They also reported that the incidence of secondary cancer in the rectal remnant after IRA was much lower than that of the previously reported 13% to 59% after 25 years of follow-up [19,20,21]. Thus, intensive endoscopic surveillance for not only the ileal pouch but also the rectal remnant, might prevent the development of secondary cancer after surgery in patients with FAP. Regarding the surveillance period, a 6-month interval in IRA patients and annually in pouch patients after surgery are recommended for the present study. However, future studies are required to determine if the interval can be extended.

Until now, the risk of pouch cancer has not been well recognized, but its existence has been gradually described and accepted. We reviewed 21 cases of ileal pouch cancers reported by 2013 [6]. Of particular note was that, in at least seven patients, the development of advanced cancer was detected within a very short interval of within 1 year after the last lower endoscopy. Although they were old reports, and their observation accuracy might have been inadequate, it may be difficult to extend the interval of lower endoscopy to one or more years until the long-term outcomes of endoscopic surveillance and treatment in pouch patients are demonstrated. In any case, the prevalence of adenoma that develops in the ileal pouch cannot be ignored, and lower endoscopy in pouch patients is needed at least once a year.

During a total of 286 treatment sessions, there were 12 cases of delayed bleeding and one of perforation, which were entirely treated by APC. Two cases of delayed bleeding required blood transfusions, and surgery was required in the case of perforation. The incidence of bleeding was significantly higher in pouch patients, and no bleeding was observed in all IRA patients. Pasquer et al. reported that the respective bleeding rates after endoscopic treatment were 16.8% in the IRA group and 15.2% in the IPAA group [18]; these rates seem to be higher compared to usual morbidity, and they attribute it to the frequent use of APC. Certainly, even in the present study, the frequency of delayed bleeding in pouch patients of 10% was considered to be much higher than that of therapeutic colonoscopy of 0.31–2.7% [22]. However, there were differences in therapeutic methods between APC in the ileal pouch and polypectomy or hot biopsy in the colorectum. Mensink et al. reported a multicenter survey of complications of double-balloon enteroscopy [23]. In their study, among 253 treatment procedures of APC, mainly for angiodysplasia, only one case (0.4%) of bleeding was reported. On the other hand, among 364 procedures of polypectomy, 12 cases (2.3%) of bleeding were reported. This difference may be in part explained by the abundant blood flow of the ileum compared to the rectum, but data to support this theory are lacking. Only one delayed perforation occurred in an IRA patient, but the perforation occurred at the ileum connected to the rectum. It was thought to have occurred when treating an adenoma that appeared in the ileal mucosa. We should be careful about perforation when we use APC for adenomas that develop in ileal mucosa, including in the ileal pouch, because the wall of the ileum is very thin.

The present study had some limitations. First, it was a single-center retrospective study with a small number of patients, and a total of 12 of the IPAA, IRA, and Kock subjects in this study were transferred to other hospitals, resulting in lost follow-up. Second, there were chronological changes or trends regarding the endoscopic diagnosis and treatment for intestinal tumors. However, this study has a maximum of 38 years of follow-up, so it cannot be avoided. Third, an *APC* gene mutation study was not performed because it was not permitted by the Japanese health insurance system. Forth, it was difficult to evaluate the histology of treated polyps using APC. However, the polyps were confirmed as adenoma before treatment, and after treatment, if there were any residual adenoma was checked by magnifying endoscopy with NBI. On the other hand, the strength of this study is that surveillance colonoscopy was performed at almost regular intervals and that indigo carmine dye endoscopy was used for careful observation, allowing for the detection of tiny adenomas in the rectum, ileal pouch, and pre-pouch. This may prevent the development of adenocarcinoma in rectal or ileal-pouch mucosa after surgical treatment in FAP patients.

## 5. Conclusions

This study demonstrated that regular endoscopic surveillance is feasible and should be recommended not only for IRA patients but also for pouch patients. Treatment for adenomas that appeared in rectal or ileal-pouch mucosa might prevent the development of adenocarcinoma. However, in pouch patients, delayed bleeding is a potential problem in the treatment of adenomas that emphasizes the need for caution.

## Figures and Tables

**Figure 1 jcm-11-03562-f001:**
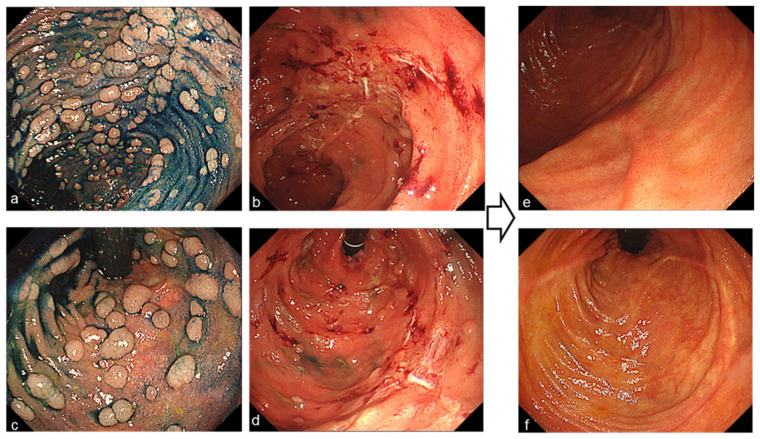
(**a**,**c**) Endoscopic image of polyposis in an ileal pouch ((**a**) oral side, (**c**) anal side) with indigo-carmine. (**b**,**d**) Endoscopic images during endoscopic mucosal resection and cold-snare polypectomy. (**e**,**f**) Endoscopic images 12 months after 2 endoscopic treatments.

**Figure 2 jcm-11-03562-f002:**
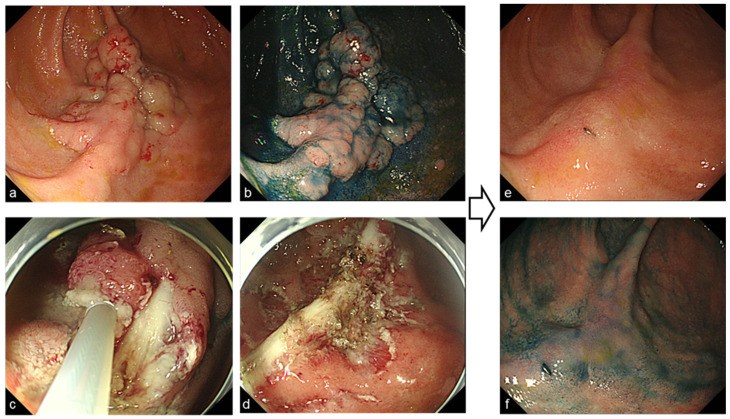
Endoscopic images before and after endoscopic piece meal resection (EPMR) for laterally spreading adenoma more than 5 cm ((**a**–**d**) before treatment, (**e**,**f**) 6 months after EPMR).

**Figure 3 jcm-11-03562-f003:**
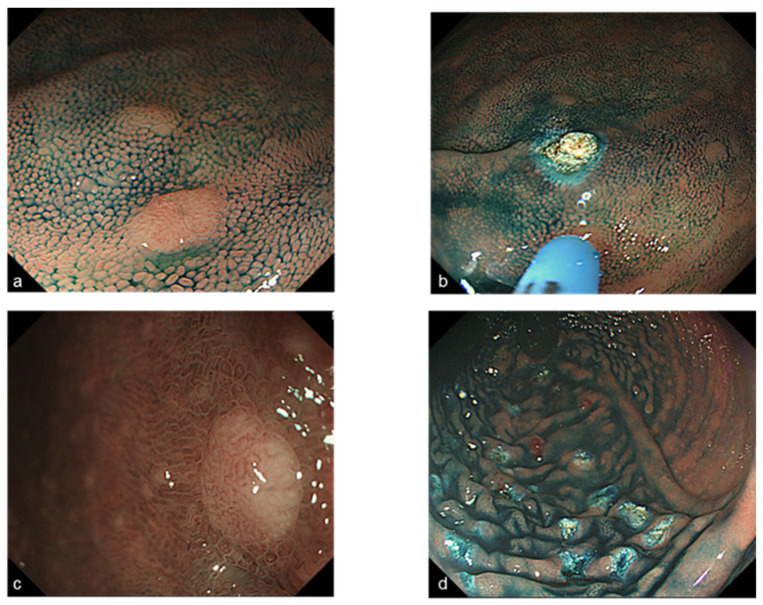
Endoscopic images before and after Argon plasma coagulation (APC) for tiny adenomas less than 5 mm (**a**) tiny polyp with indigo-carmine, (**c**) magnifying endoscopy image with narrow-band imaging shows JNET Type 2A (diagnosed as adenoma), (**b**,**d**) Endoscopic images after APC.

**Figure 4 jcm-11-03562-f004:**
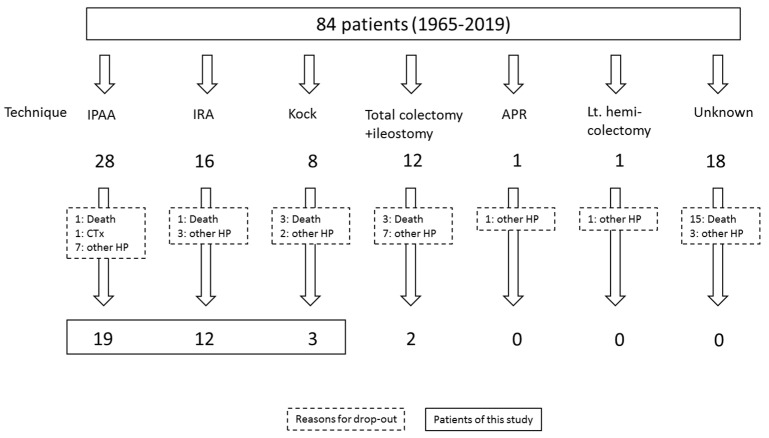
Patient flow in this study.

**Table 1 jcm-11-03562-t001:** Characteristics of pouch patients and IRA patients and details of endoscopic surveillance.

	Pouch Patients	IRA Patients	*p*-Value
*n*	22	12	
Male, *n* (%)	11 (50)	4 (33.3)	0.566
Median age at surgery, y (range)	30.6 (17–52)	36.7 (19–67)	0.098
Median polyp count at surgery			
Total	2245 (600–9838)	1339 (200–9436)	0.596
Colon	1968 (500–17,200)	1399 (150–9340)	0.608
Rectum	95 (5–5282)	73 (1–607)	0.197
Gastric polyp, *n* (%)	17 (77.2%)	11 (91.7%)	0.561
Papillary adenoma, *n* (%)	10 (45.4%)	4 (33.3%)	0.748
Extra-papillary adenoma, *n* (%)	8 (36.3%)	5 (41.6%)	0.948
Median follow-up period since surgery, y (range)	24.3 (3.7–40.2)	21.7 (1–39.7)	0.441
Median interval to 1st endoscopy after surgery, y	8.0 (0.9–21.9)	0.5 (0.4–1.8)	<0.001
(range)			
Frequencies of adenoma development, *n* (%)	20 (90.9)	11 (91.6)	0.577
Median duration to 1st detection of adenoma after surgery, y (range)	11.7 (1.0–30.2)	1.3 (0.5–6.7)	<0.001

**Table 2 jcm-11-03562-t002:** Clinical outcomes of endoscopic surveillance.

	Pouch Patients	IRA Patients	*p*-Value
*n*	22	12	
Median follow-up period from the start of surveillance, (y) median (range)	11.3 (2.2–17.1)	21.7 (1.0–38.0)	0.202
Interval of lower endoscopy, (months) median (range)	12 (6–24)	6 (6–24)	0.097
Number of lower endoscopies, *n*, median (range)	13.5 (2–50)	18.0 (2–60)	0.208
Number of endoscopic treatments per patient, *n*, median (range)	5.0 (1–16)	13.5 (1–48)	0.074
Number of treated adenomas per treatment, *n*, median (range)	12.5 (3–150)	13.5 (3–60)	0.454
Cumulative number of treated adenomas per patient, *n*, median (range)	55 (6–480)	172.5 (20–1106)	0.191
Cumulative number of treatments (all patients)	120	169	
Detail of procedures			
APC, *n* (%)	116 (96.7)	162 (95.6)	
EMR & EPMR, *n* (%)	8 (6.7)	5 (3.0)	0.007
Cold polypectomy, *n* (%)	15 (12.5)	4 (2.4)	
Combination, *n* (%)	9 (7.5)	7 (4.1)	
Total number of cumulative polyps	2132	2682	
Secondary cancer	0	0	
Additional surgery	0	0	
Complications			
Delayed bleeding: % (*n*/N)	10 (12/120)	0 (0/169)	<0.001
Perforation: % (*n*/N)	0 (0/120)	0.6 (1/169)	0.863

APC, argon plasma coagulation; EMR, endoscopic mucosal resection; EPMR, endoscopic piecemeal mucosal resection.

## Data Availability

All data generated or analyzed during this study are included in this article. Further enquiries can be directed to the corresponding author.

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
