# Peer review of "Endoscopic Management of Adenomas in the Ileal Pouch and the Rectal Remnant after Surgical Treatment in Familial Adenomatous Polyposis"

_jcm, 2022, doi:10.3390/jcm11123562_

Round 1

Reviewer 1 Report

I think it's a very interesting study. Given that this condition is not very common, the small study cohort is accepted. However, the long period of time in which these patients were followed and the evolution of endoscopic techniques may lead to some errors in the reporting of results. 
I also consider it necessary to better analyze the literature to compare the results of this study.

Author Response

Response: Thank you very much for your constructive comments. As you mentioned, this study is a long-term retrospective study, and it is undeniable that advances in endoscopic diagnosis and treatment during that period may have influenced the results of the study. So, this point is mentioned in the second limitation (L282-284). Furthermore, as you mentioned, I want to compare our study with published study. However, there were quite few reports as to treatment for ileal pouch tumor in patients with FAP. As much as possible, I mentioned the comparison with the past reports in the discussion of Line 257-273.

Reviewer 2 Report

I thought this was a nice article.  It was very helpful to see the endoscopic images.  A few questions:

1. Why the high wattage for APC in the small bowel.  I am typically using lower wattage.  

2.  When there was delayed bleeding was it with the older models that do not have the ability to energy adjust like the 300d?  If the delayed bleeding happened with the older models it should be noted.

3. Why was there a perforation at the ileal-rectal anastamosis?

Author Response

I thought this was a nice article.  It was very helpful to see the endoscopic images.  A few questions:

  1. Why the high wattage for APC in the small bowel.  I am typically using lower wattage.  

Response: Thank you very much for your useful advice. However, in this study, we used APC for the purpose of eradication of adenomas. We think that 30 watts is necessary even in the small intestine. As you mentioned, we think it is necessary to be careful when using APC in the small intestine, so we are calling for caution. (L276-278)

  1. When there was delayed bleeding was it with the older models that do not have the ability to energy adjust like the 300d?  If the delayed bleeding happened with the older models it should be noted.

Response: Thank you very much for your useful advice. However, it is difficult to prove that the cause is the performance of the machine. Regarding this point, we will not describe it because there is no data to compare.

  1. Why was there a perforation at the ileal-rectal anastamosis?

Response: A perforation had not occurred at the ileal-rectal anastomosis but at the ileal mucosa connected to the rectum. The expression was difficult to understand, so we changed the description as follows. L273-276